# The Influence of Rapid Solidification on Corrosion Behavior of Mg_60_Zn_20_Yb_15.7_Ca_2.6_Sr_1.7_ Alloy for Medical Applications

**DOI:** 10.3390/ma14195703

**Published:** 2021-09-30

**Authors:** Katarzyna Młynarek-Żak, Anna Sypien, Tomasz Czeppe, Anna Bajorek, Aneta Kania, Rafał Babilas

**Affiliations:** 1Department of Engineering Materials and Biomaterials, Silesian University of Technology, Konarskiego 18a, 44-100 Gliwice, Poland; aneta.kania@polsl.pl (A.K.); rafal.babilas@polsl.pl (R.B.); 2Institute of Metallurgy and Materials Science of Polish Academy of Sciences, 25 Reymonta 5 St., 30-059 Kraków, Poland; a.sypien@imim.pl (A.S.); t.czeppe@imim.pl (T.C.); 3A. Chelkowski Institute of Physics, University of Silesia in Katowice, 75 Pułku Piechoty 1, 41-500 Chorzów, Poland; anna.bajorek@us.edu.pl

**Keywords:** Mg-based alloys, scanning electron microscopy, energy-dispersive X-ray spectrometer, differential scanning calorimetry, corrosion resistance, X-ray photoelectron spectroscopy, hardness tests

## Abstract

Biodegradable magnesium alloys with Zn, Yb, Ca and Sr additions are potential materials with increased corrosion resistance in physiological fluids that ensure a controlled resorption process in the human body. This article presents the influence of the use of a high cooling rate on the corrosion behavior of Mg_60_Zn_20_Yb_15.7_Ca_2.6_Sr_1.7_ alloy proposed for medical applications. The microstructure of the alloy in a form of high-pressure die-casted plates was presented using scanning electron microscopy in the backscattered electrons (BSEs) mode with energy-dispersive X-ray spectrometer (EDX) qualitative analysis of chemical composition. The crystallization mechanism and thermal properties were described on the basis of differential scanning calorimetry (DSC) results. The corrosion behavior of Mg_60_Zn_20_Yb_15.7_Ca_2.6_Sr_1.7_ alloy was analyzed by electrochemical studies with open circuit potential (E_OCP_) measurements and polarization tests. Moreover, light microscopy and X-ray photoelectron spectroscopy were used to characterize the corrosion products formed on the surface of studied samples. On the basis of the results, the influence of the cooling rate on the improvement in the corrosion resistance was proved. The presented studies are novel and important from the point of view of the impact of the technology of biodegradable materials on corrosion products that come into direct contact with the tissue environment.

## 1. Introduction

Magnesium fulfills a very important role in the functioning of living organisms. This element plays an essential role in the formation of antibodies and in the blood clotting process. In the skeletal system, Mg is responsible for the proper absorption of calcium, activates the osteosis process, improves bone tissue density, and stimulates the hormone that triggers the release of calcium from the blood and deposition in the bones [1]. Pure magnesium possesses mechanical properties close to human bone, compared to clinically used materials [2]. Therefore, magnesium alloys can be used as load-bearing orthopedic implants that remain in the body and maintain mechanical integrity for 12 to 24 weeks [3]. During this time bone heals and is finally replaced by natural tissue [3,4,5]. However, the main limitation of the wider usage of Mg alloys in the human body is the uncontrolled corrosion process that leads to premature failure of the implant.

The addition of alloying elements is a proper and practicable measure to address the high corrosion resistance and hardness. The ideal composition of the alloying elements in Mg alloys would stabilize the hydroxide film (which plays an essential role in corrosion resistance) on the surface of these alloys, increase their mechanical properties, and thus improve their biocompatibility [5,6].

Determination of appropriate casting methods for magnesium alloys is also important to obtain desired properties (i.e., improved corrosion resistance, good mechanical properties, etc.). In practice, there are many casting methods to choose from, for example, gravity casting (GC), high-pressure die casting (HPDC), squeeze casting, semi-solid metal casting, and other casting processes [7,8]. The most commonly used casting process for Mg alloys is the HPDC method. In this method, molten metal is injected into the mold at high speed and pressure, and solidifies quickly under these conditions, which includes a precision casting process. High-pressure die casting to water-cooled copper mold provides an increased cooling rate, compared to gravity casting. It should be mentioned that the parameters of the casting process (i.e., cooling rate) affect the Mg alloy microstructure, and some changes in the microstructure affect its mechanical, and also corrosion properties [9]. During the solidification process, a faster cooling rate may result in a reduction in dislocation accumulation [7]. Therefore, HPDC is the main process used to produce Mg alloys with high efficiency, reduced mechanical post-treatment, and fast cooling, which can effectively refine the α-Mg main matrix and improve the mechanical properties of the alloys. Shastri et al. noted that AZ91 alloy produced by HPDC has better mechanical properties, compared with the gravity casting [10]. Both the strength of the alloys and corrosion resistance are of significant importance in casting technology. However, in the literature, there is still a lack of studies on the effects of casting parameters (i.e., gravity casting and HPDC) on the corrosion behavior in Ringer’s solution. However, the grain refinement in the Mg alloys prepared by high-pressure die casting has been found to improve the corrosion resistance by reducing the pitting initiation [11]. Therefore, the authors made an attempt to determine the effect of rapid solidification on the corrosion resistance of the selected Mg_60_Zn_20_Ca_2.6_Yb_15.7_Sr_1.7_ alloy in a chloride-rich solution.

Many studies point out that important alloys among biodegradable Mg implants are Mg-Zn-Ca, fabricated in many casting processes. They contain elements naturally occurring in the human body that can be metabolized and released in a natural way [12]. Biodegradable Mg-Zn-Ca alloys are used in medicine for short-term orthopedic implants [13]. Zinc, as an alloying element, improves corrosion resistance and also reduces grain size [5,14]. Calcium added to magnesium alloys improves mechanical properties at room temperature and increases hardness and resorbability.

The influence of the alloying elements on the microstructure, corrosion behavior, and biocompatibility of Mg-based alloys is widely studied based on specific aspects of the physiological environment, the electrochemical effect, and biological behavior [15]. Strontium, similar to calcium, is incorporated at a similar rate into bone cells (increasing their density), thereby preventing the development of osteopenia and osteoporosis [16,17]. Ytterbium, a rare earth element, has an effect on refining the microstructure and improvement in the corrosion resistance of Mg alloys [18,19]. In the literature, there is no information on the optimal content of Yb in Mg alloys from the corrosion point of view, although the solubility limit of Yb in Mg-based alloys is 4.8% at 400 °C. Li et al. [18] studied biocorrosion behavior and mechanical properties of ZK60 alloy with 0, 1, and 2 Yb wt.%. They argued that by increasing the ytterbium content, the grain size was decreased. The refined microstructure also has a positive effect on the increase in microhardness. The corrosion behavior was investigated by electrochemical and immersion tests carried out in simulated body fluid (SBF) solution. In both cases, with an increase in Yb content, the corrosion resistance was improved. In the Mg alloys with ytterbium, a higher corrosion potential was observed in combination with a lower corrosion current density and pH value [18]. Yamasaki et al. [19] investigated the Mg-Zn-Yb and Mg-Zn-La alloy ribbons in 1% NaCl solution to determine the optimum composition of corrosion-resistant Mg alloys. The researchers confirmed that Mg-Zn-Yb alloy exhibited a low corrosion rate in saline solution. In other published works, it was stated that the addition of ytterbium at the level of 2 wt.% also improves bending plasticity [13,20].

Due to the biocompatibility and anticorrosive properties, strontium was selected to prepare Mg-Sr alloy (with the addition of Sr from 1 to 4 wt.%) [21,22,23]. Gu et al. [21] investigated the optimal Sr content, taking into account the mechanical and corrosion resistance of the Mg-Sr alloys. In addition, the possibility of using the Mg-Sr alloys as orthopedic implants has been studied by in vitro cellular experiments and intramedullary implantation tests. The authors confirmed that the optimal Sr content in Mg-based alloys was 2 wt.%. Mg-2Sr alloy exhibits the highest corrosion resistance and strength. Four weeks after the implementation, increased mineral density and a thicker cortical bone around the experimental implants were observed [21]. In a study by Zhao [22], electrochemical and immersion tests carried out in Hank’s solution showed that the Mg-0.5Sr alloy exhibit the best anticorrosion properties, and the anticorrosion properties decreased with increasing Sr content. The cytotoxicity tests also showed that the Mg-0.5Sr alloy did not induce cell toxicity. Moreover, Bornapour et al. [23] studied the combined effect of calcium and strontium on the corrosion behavior of Mg-based alloys. The electrochemical and immersion tests were realized in the SBF solution. During corrosion tests, the changes in mechanical properties were investigated. The authors stated that the addition of 0.3 wt.% Sr and 0.4 wt.% Ca into Mg alloys decreased the corrosion rate both in electrochemical and immersion tests (a hydrogen evolution rate in SBF solution was 0.01 mL·cm^−^^2^·h^−^^1^). Higher concentrations of Sr and Ca lead to an increase in the corrosion rate by promoting microgalvanic corrosion [23]. The in vitro and in vivo corrosion resistance analyses of two ternary Mg-2Sr-Zn and Mg-2Sr-Ca alloys were conducted by Chen et al. [24]. The corrosion resistance of both studied alloys was improved in comparison with previous Mg-2Sr alloys. The degradation rate was 1.1 and 0.85 mm·y^−^^1^ for the Mg-2Sr-Ca and Mg-2Sr-Zn alloys, respectively (the corrosion rate was 1.37 mm·y^−^^1^ for Mg-2Sr alloy). Other works also state that the addition of strontium less than 2 wt.% decreased the corrosion rate and changed the mechanical properties of magnesium alloys to values close to the strength of bone tissue [20,25,26]. Li et al. [27] confirmed that the addition of Sr (less than 1.5 wt.%) into MgCaZn alloy significantly improves the corrosion resistance and mechanical properties. Moreover, it was found out that the addition of Zn to the Mg-Sr-Ca alloy promotes the growth of osteoblasts and improves antibacterial properties [28].

The aim of the work was to investigate the corrosion behavior of Mg_60_Zn_20_Ca_2.6_Yb_15.7_Sr_1.7_ alloy in a form of slowly cooled ingots and plates obtained by HPDC. Our results are discussed in the context of structural investigations, corrosion behavior, and microhardness of Mg_60_Zn_20_Ca_2.6_Yb_15.7_Sr_1.7_ alloy, and they could be very important in determining the nature of future generations of biodegradable orthopedic implants.

## 2. Materials and Methods

### 2.1. Preparation of the Samples

The chemical composition of Mg_60_Zn_20_Ca_2.6_Yb_15.7_Sr_1.7_ alloy was selected taking into account the optimization of chemical compositions presented in [20]. Biocompatible elements with the purity of 99.9% and a form of pieces were weighed and melted by using induction generator NG-40F (Łukasiewicz Research Network—Institute of Welding, Gliwice, Poland). Master alloys in the form of ingots were prepared by gravity casting into ceramic (Al_2_O_3_) crucible. The plates were prepared by the HPDC method in a water-cooled copper mold under argon protection, which resulted in an increased cooling rate. The casting process of ingots and plates was conducted with argon (99.999%) atmosphere to provide air reduction.

### 2.2. Structural and Calorimetric Investigations

Observations of the rapidly solidified alloys were performed using scanning electron microscopy in backscattered electrons (BSEs) mode (SEM, FEI Quanta 3D FEGSEM, FEI, Hillsboro, OR, USA) with energy-dispersive X-ray spectrometer (EDX) analysis to identify the chemical composition of visible phases. The mechanism of crystallization of ingots and plates was analyzed on the basis of recorded differential scanning calorimetry (DSC, 910 model, DuPont Company, Wilmington, DE, USA) curves during heating with 30°/min. Additionally, the heating and cooling process for the alloy in a form of a plate was performed in a higher temperature range of 350–875 °C with a reduced rate to 20°/min by thermal analyzer SDT Q600 (TA Instruments, New Castle, DE, USA).

### 2.3. Corrosion Studies

Corrosion resistance was assessed on the basis of electrochemical tests. The measurements were carried out in a Ringer’s solution at 37 °C using an Autolab 302N potentiostat (Metrohm AG, Herisau, Switzerland). The conditions of the experiment were similar to the natural environment inside the organism. The potentiostat was equipped with a cell containing the reference electrode (saturated calomel electrode) and the counter electrode (platinum rod). Mg_60_Zn_20_Ca_2.6_Yb_15.7_Sr_1.7_ alloy in a form of ingots and plates were tested by 3600 s of open-circuit potential (E_OCP_) at a scan rate of 1 mV s^−1^. The polarization curves with Tafel’s extrapolation were determined after stabilization time.

The corrosion products were assessed on the basis of light microscopic images. Additionally, XPS analysis of the samples after electrochemical tests was used.

X-ray photoelectron spectroscopy (XPS) measurements of ingot and plate samples after corrosion tests in Ringer’s solution at 37 °C were conducted in PHI 5700/660 Physical Electronics spectrometer under the high vacuum of 10^−10^ Torr. The monochromatic Al K_α_ (1486.6 eV) X-ray radiation was used. The depth-profile (DP-XPS) analysis was performed in sputtering cycles by using Ar^+^ beam of 1.5 kV applied for 15 min. After each argon beam etching, the valence bands, as well as the core level lines mostly for the elements with the highest photoemission cross section were acquired. The total sputtering time for studied samples was about 300 min. All the collected spectra were determined relative to the C1s peak 284.8 eV, which was used as a reference for charge correction. The data analysis was made by using MultiPak 9.8 software. The chemical states and chemical shifts were assigned with the help of the MultiPak internal database and NIST XPS database. The survey spectra were collected with the pass energy 187.85 eV. The core level lines were measured with the pass energy of 23.50 eV and the resolution of 0.1 eV.

### 2.4. Mechanical Tests

The basic mechanical properties were investigated by hardness tests. The Vickers method was used for the measurements. The studies were performed with the Future-Tech FM-ARS (9000 model) device under the load of 100 gf and with a duration time of 15 s. For the studied alloy, 10 measurements were carried out. The average microhardness value with the standard deviation was calculated.

## 3. Results and Discussion

### 3.1. Structural and Calorimetric Investigations

The microstructure of Mg_60_Zn_20_Ca_2.6_Yb_15.7_Sr_1.7_ alloy in a form of an HPDC plate is shown in Figure 1 as an image recorded in the BSE mode with the chemical elements’ qualitative analysis from the selected area. The microstructure of the studied alloy had multi-phase nature, characterized by a dendritic structure (Figure 1a,b). The structural images of Mg-Ca-Sr-(Zn) alloys were presented in the publication [28]. In the article, all studied chemical compositions were characterized by a similar microstructure as Mg_60_Zn_20_Ca_2.6_Yb_15.7_Sr_1.7_ alloy in a form of plate, in which the images were presented with lower magnification. The authors of the publication [28] described the microstructure as composed of secondary phases distributed along the grain boundaries and black particles, the presence of which increased with the higher zinc content. The qualitative analysis confirmed the presence of Zn, Yb, Ca, and Sr alloying elements (Figure 1c).

The DSC curves were recorded in order to characterize the thermal properties and the crystallization mechanism of Mg_60_Zn_20_Ca_2.6_Yb_15.7_Sr_1.7_ alloy in a form of ingots and plates, in the temperature range of 200–500 °C with a heating rate of 30°/min (Figure 2). The occurrence of one endothermic thermal effect with the onset around 440 °C was observed for both kinds of samples. The heating and cooling curves in a larger temperature range of 350–875 °C with reduced heating and cooling rate to 20°/min were recorded for the HPDC plate due to the similarity of the heating curves presented in Figure 3. The heating curve was characterized by two overlapping endothermic thermal effects with an onset of the melting point of 446 °C. The occurrence of an exothermic effect with an initial temperature of 676 °C was also observed. In the case of the cooling curve, there were exothermic effects corresponding to the phase transformations for heating in the range of 400 °C. In the literature, the measurements with the use of differential scanning calorimetry for Mg-Zn-Ca-(Sr) alloys were recorded in the range of 350–700 °C [27,29].

### 3.2. Corrosion Behavior

Magnesium, similar to most metals and alloys, possesses a natural passive surface layer that controls its corrosion behavior. A surface layer consists mainly of Mg(OH)_2_, with probably MgO at the metal film interface [30,31]. The structure and composition of the surface film strongly depend on metallurgical and environmental factors (i.e., impurities in the metal and electrolyte species, such as Cl^−^ ions), which determine the protective ability of the passive film.

It should be mentioned that the oxide films formed on the surface of Mg-based alloys are porous. Moreover, the corrosion rate increases with the increase in Cl^−^ ions concentration, and the chloride attack usually results in pitting corrosion.

In the present work, the corrosion behavior was evaluated from the open circuit potential (E_OCP_) and Tafel’s extrapolation of polarization curves presented in Figure 4. Moreover, the calculated values of E_OCP_, corrosion potential (E_corr_), anodic and cathodic Tafel slopes (β_a_, β_c_), polarization resistance (R_p_), corrosion current density (j_corr_) are listed in Table 1. The measurements of E_OCP_ changes were used to determine the protective properties of films formed on the surface in Ringer’s solution at 37 °C. The curves determined for a stationary potential as a function of time (equals 3600 s) indicated that the studied samples were active in a chloride-rich environment. The clear effect of the applied cooling rate on the improvement in corrosion resistance is visible by the shift toward the positive values of E_OCP_ and E_corr_ potentials, increased value of the polarization resistance, and reduction in the corrosion current density (Table 1). In the case of the chemical composition, the measurement results were compared with the literature [15,20,32]. According to the publication [15], the addition of Sr ≤ 2% reduces the corrosion rate of Mg-Zr alloys and pure magnesium. The article [20] underlined that the addition of Sr and Yb improves the corrosion resistance of calcium alloys. Moreover, the authors mentioned that the addition of 2% of Yb improves the properties of Mg alloys, including corrosion resistance [20]. In their work, Datta et al. [32] compared the polarization curves recorded for pure magnesium and Mg_60_Zn_35_Ca_5_ alloy in a form of CIP (cold isostatic pressing) compacted pellet consisting of amorphous structure and sintered pellet in Dulbecco’s modified Eagle medium. The alloy with the addition of Zn and Ca showed an improvement in corrosion resistance, compared to pure magnesium [32]. In relation to the results in this study, the E_corr_ values of Mg_60_Zn_20_Ca_2.6_Yb_15.7_Sr_1.7_ alloy showed a significant improvement in relation to pure Mg [32]. In the case of the Mg_60_Zn_35_Ca_5_ alloy produced by two different technologies, the values of the corrosion potential of the alloy with the addition of Yb and Sr showed more negative values [32]. However, in this article, more favorable corrosion current density values were achieved. The values were, respectively, for Mg_60_Zn_35_Ca_5_ in the form of a CIP compacted pellet consisting of an amorphous structure (21.38 mA·cm^−2^) and sintered pellet (12 mA·cm^−2^) [32]. Another article [29] described the results of electrochemical tests in SBF solution for the Mg_65.2_Zn_28.8_Ca_6_ alloy in the form of ingots, copper-mold-casted specimens with an amorphous structure, and micro-arc oxidation (MAO) treated rapidly solidified alloy. In the case of the value of the corrosion potential, the studied ingots were characterized by a value similar to slowly cooled alloys (−1.475 ± 0.029 V) analyzed in the study [29]. Similar results for the value of the corrosion potential were also shown for rapidly solidified alloys because the value presented in the publication was −1.345 ± 0.031 V [29].

After the electrochemical tests, the samples of the Mg_60_Zn_20_Ca_2.6_Yb_15.7_Sr_1.7_ alloy were observed by stereoscopic microscope (Figure 5). The surfaces of the alloy in both forms of ingots and plates presented the post corrosion damages. We can also conclude that the corrosion product layers are dense. Microcracks and small pits are observed on the surface of the studied samples, revealing that these alloys undergo pitting corrosion [31].

Figure 6 demonstrates the XPS survey spectra for the ingot and plate of Mg_60_Zn_20_Ca_2.6_Yb_15.7_Sr_1.7_ samples after corrosion tests in Ringer’s solution at 37 °C. Both spectra were recorded for the surface before applying the argon beam. Therefore, some photoemission states (especially for ingot samples) are barely noticeable due to covering by surface impurities and oxides, which subsequently are removed during sputtering. However, one may easily notice the main photoemission lines such as Mg1s, Mg2p, O1s, C1s, Yb4d, and Sr3d, as well as Auger peaks such as C KLL, O KLL, and the set of Mg KLL.

The analysis of individual core level lines upon sputtering rate is presented in Figure 7 and Figure 8 for the ingot and plate form of a sample, respectively. Note that each core level line acquired during collecting DP-XPS is depicted as normalized stacking plots. In each of them, the surface components are placed on the bottom, whereas those collected at the end of the sputtering process are placed at the top.

The oxygen line O1s (see Figure 7a and Figure 8a) for both samples is rather complex. On the surface, it is more likely dominated by the MgO (BE = 531.8 eV) states probably overlapped with Mg(OH)_2_. The presence of Na_2_(CO)_3_ (BE = 531.7 eV) states, as an effect of Ringer’s solution residual, is not excluded, especially that a very small Na1s peak (BE ≈ 1071 eV) can be noticeable in the surface survey spectra. During the argon etching process, the O1s line becomes broader, leading to a more complex structure. The appearance of the peak around BE H 528 ÷ 529 eV is consistent with the emergence of Zn and Yb states, which were covered by surface components. Thus, such a peak is typical for various metallic oxides (probably mostly Yb_2_O_3_), which, in this case, are overlapped with Mg states. The influence of Ca and Sr elements for the O1s line is rather minimal due to their low quantity in the sample formula.

The Mg2p core level lines (see Figure 7b and Figure 8b) varies significantly with the sputtering rate. Initially, on the surface, we can observe a broad spectrum (especially for ingot specimen), which is probably dominated mostly by the MgO component (BE ≈ 50.1 eV) partially overlapped with Mg(OH)_2_ states. The applying of the argon beam revealed the emergence of a low binging energy peak (BE ≈ 48 ÷ 47 eV), which can be assigned to a pure magnesium state.

The Zn2p surface states are depicted in Figure 7c and Figure 8c for ingot and plate samples, respectively. They are barely visible and dominated by ZnO components. After surface cleaning, the zinc line becomes narrow and is composed of two species Zn2p_3/2_ (BE ≈ 1019 eV) and Zn2p_1/2_ (BE ≈ 1042 eV) with a spin-orbit splitting of ∆E ≈ 23 eV, which is rather typical for zinc states. The narrower zinc lines and their binding energies can be assigned to pure Zn.

The Ca2s line (see Figure 7d and Figure 8d) is very weak due to the low quantity of calcium in the sample formula and a quite small photoemission cross section for this line. However, one may barely detect some broad components mostly associated with CaO states.

The Yb4d states at the surface (see Figure 7e and Figure 8e) reveal a doublet line with L-S splitting into two main lines, Yb4d_5/2_ (BE ≈ 186 eV) and Yb4d_3/2_ (BE ≈ 199 eV), with LS splitting of about ∆E ≈ 13 eV. Both surface states are associated with Yb_2_O_3_. The application of the Ar+ beam leads to the appearance of additional components at lower binding energies, which can be assigned as pure Yb states. In addition, as one may observe, Yb4d component peaks are broad, and there are additional structures, e.g., around 193 eV. The complex nature of the Yb4d ytterbium structure is typical for mixed-valence Yb^3+^/Yb^2+^ ytterbium states. Here, it seems that such a structure is present both on the surface and also in the deeper layers, where in addition pure ytterbium states are revealed.

The Sr3d lines (see Figure 7f and Figure 8f), both on the surface and in the interior of the sample, reveal a dual nature. Thus, the presence of SrO and Sr states can be distinguished. The first mentioned ones are noticed for higher binging energy range peaking around BE ≈ 137 eV, whereas the latter ones around BE ≈ 133 eV. Obviously, both peaks assigned to strontium oxides and pure strontium are broad due to the presence of Sr3d_5/2_ and Sr3d_3/2_ states with narrow spin-orbit splitting of about ∆E ≈ 1.8 eV for each species.

The depth profiles based on the individual core level lines over ion etching by argon beam are presented in Figure 9 for the ingot and plate samples. Obviously, at the surface dominates impurities such as C1s which are eradicated over ion beam cleaning up to the level of less than 2–3 at.% after the last stage of ion etching—namely, 300 min. Notably, the quantity of O1s is different by about 10 at.% in both samples. For ingot at the surface, it is about 20 at.%, while for plate is about 30 at.%. In addition, the level of O1s is reduced faster for the ingot sample (up to 7 at.%) than for the plate (up to about 21 at.%). Admittedly, such a difference is probably associated with the higher O1s accumulated at the surface as an impurity for ingot, whereas for the plate sample, more O1s can be assigned as various oxides or even hydroxide states present in the interior of the specimen. Indeed, for the plate sample, we observe a higher level of Yb4d (up to about 35.1 at.%), Mg2p (up to about 24.8 at.%), Zn2p (up to about 12.1 at.%), and Sr3d (up to about 3.8 at.%). Additionally, probably in the plate sample, we have more pure Yb than in its ingot counterpart.

Such a different atomic composition observed for both samples under the same DP-XPS measurement conditions may be related to the different chemical states. It is also worth mentioning that XPS is a surface-sensitive method, and the quality of the surface (e.g., roughness, surface impurities, etc.) plays a significant role in the determination of percentage atomic composition. Nonetheless, it is also unambiguous to determine the exact chemical composition on the surface and upon sputtering due to the possible interdiffusion process induced by Ar^+^ beam, which is probably more likely to be realized for a more porous sample.

### 3.3. Mechanical Properties

In order to investigate the basic mechanical properties of the Mg_60_Zn_20_Ca_2.6_Yb_15.7_Sr_1.7_ alloy in a form of a plate, the hardness tests by the Vickers method under a load of 100 gf (0.98 N) were carried out. The average value for 10 measurements was 297 ± 4 HV. The obtained value of microhardness for the studied alloy is much higher, compared with the Mg-based alloys with ytterbium addition presented in the work [18]. In the studies of Li et al. [18], the microhardness values obtained under the load of 9.8 N for the Mg_5.8_Zn_0.5_Zr_x_Yb (x = 0, 1.0, and 2.0) alloys were not higher than 81 HV. It should be mentioned that in the other works, the microhardness values measured with the same load conditions as in our studies were lower, compared with the studied alloy, and equaled 57 ÷ 58 HV for AZ91D [33], 46 ± 1 HV for Mg-0.6Ca, 47 HV for Mg-1Ca [33], and Mg-0.5Ca-0.3Zn alloys [34]. Riaz et al. [35] showed that the addition of calcium, zinc, and rare earth metals (i.e., Yb) into Mg alloys increases the hardness, which was confirmed by the results presented in this work. The refined microstructure of the Mg alloys with ytterbium addition has a positive effect on the increase in microhardness values [35].

## 4. Conclusions

In this article, the corrosion behavior of new Mg_60_Zn_20_Ca_2.6_Yb_15.7_Sr_1.7_ alloy in a form of ingots and plates was studied. Additionally, the impact of two different cooling rates (casting methods) on the corrosion resistance of the studied alloy was analyzed. On the basis of the obtained results of corrosion tests, it was stated that the Mg_60_Zn_20_Ca_2.6_Yb_15.7_Sr_1.7_ alloy produced by the HPDC method was characterized by improved corrosion resistance, compared with the second one. This was visible by the higher values of open circuit potential, corrosion potential, polarization resistance, and lower values of corrosion current density (Table 1). The pitting corrosion mechanism was assessed on the basis of microscopic observation of the surface morphology of the studied alloy after electrochemical tests. Small pits and microcracks were observed in both forms (ingot and plate) of the Mg_60_Zn_20_Ca_2.6_Yb_15.7_Sr_1.7_ alloy.

Based on the XPS core level lines spectra, it was observed that the surface of the ingot was characterized by a higher presence of oxides, especially MgO. It was also found that for all the alloying elements, except calcium, due to its low content, there were bound to oxygen on the surface, while no oxides were observed after the application of an argon beam. On the basis of DP-XPS results, the difference in O1s was observed. This was resulted from impurities on the surface of the ingot, while in the plate, oxygen was associated with Yb, Mg, Zn, and Sr.

The basic mechanical properties of the plate form of the studied alloy were analyzed by hardness tests. The obtained value of Vickers hardness for the Mg_60_Zn_20_Ca_2.6_Yb_15.7_Sr_1.7_ alloy was 297 ± 4 HV. This value of microhardness was much higher than other data referred to in the literature for the same leading conditions.

## Figures and Tables

**Figure 1 materials-14-05703-f001:**
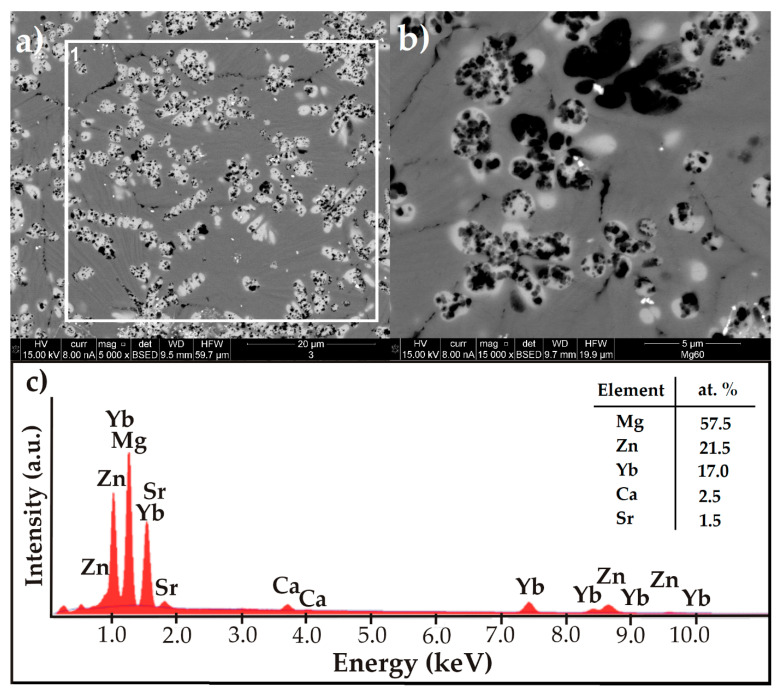
SEM-BSE image of Mg_60_Zn_20_Ca_2.6_Yb_15.7_Sr_1.7_ alloy in a form of a plate (**a**,**b**) and EDX spectra (**c**) of the selected area (frame 1).

**Figure 2 materials-14-05703-f002:**
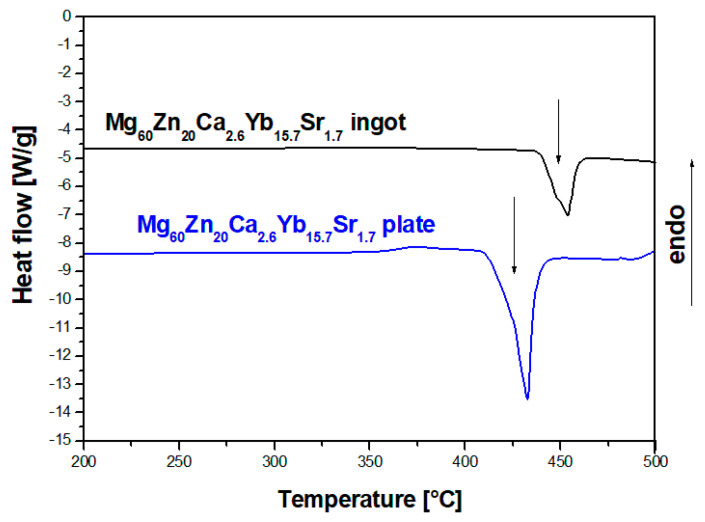
DSC heating curves of Mg_60_Zn_20_Ca_2.6_Yb_15.7_Sr_1.7_ alloy in a form of ingot and plate, 30°/min.

**Figure 3 materials-14-05703-f003:**
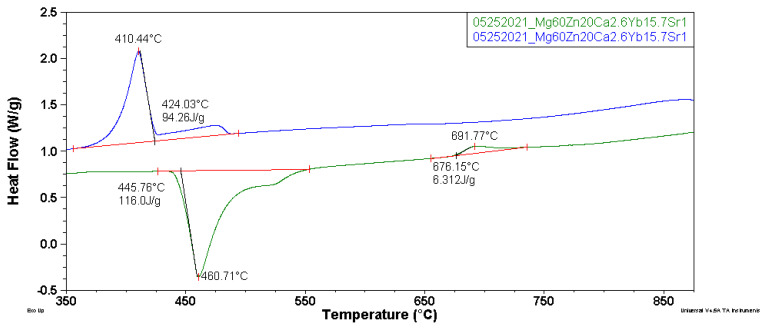
DSC heating and cooling curves of Mg_60_Zn_20_Ca_2.6_Yb_15.7_Sr_1.7_ alloy in a form of plate, 20°/min.

**Figure 4 materials-14-05703-f004:**
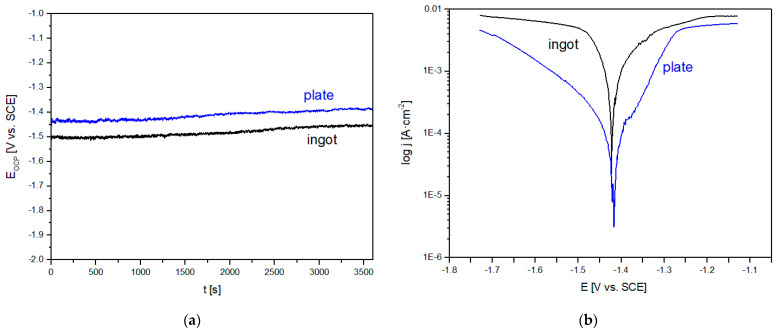
Changes of an open-circuit potential with time (**a**) and polarization curves (**b**) of Mg_60_Zn_20_Ca_2.6_Yb_15.7_Sr_1.7_ alloy in a form of ingots and plates.

**Figure 5 materials-14-05703-f005:**
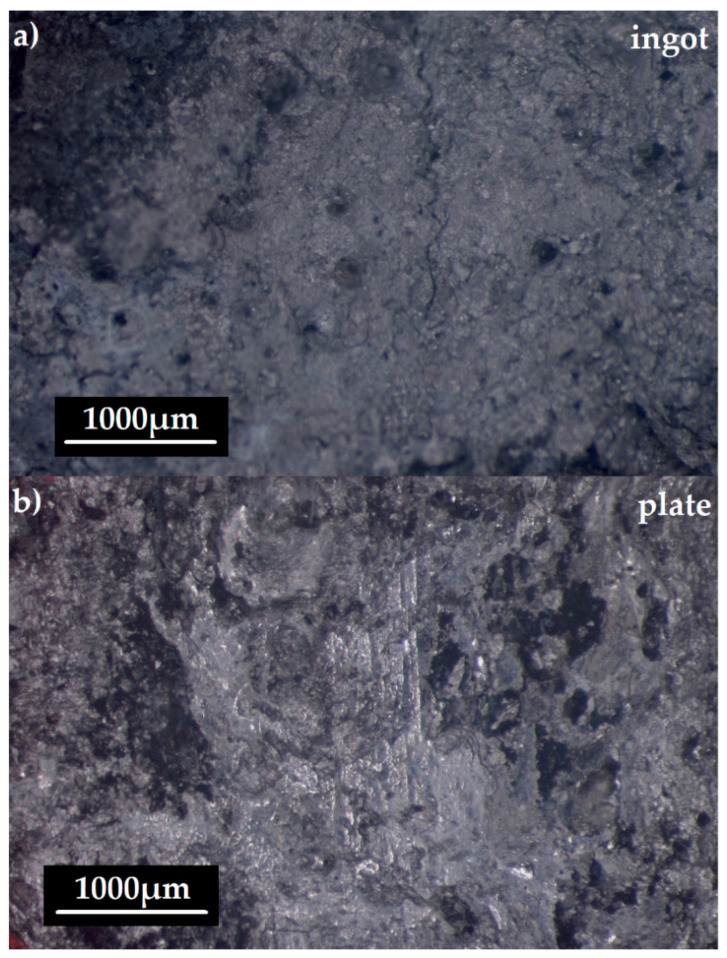
Surface morphology after electrochemical tests of Mg_60_Zn_20_Ca_2.6_Yb_15.7_Sr_1.7_ alloy in a form of (**a**) ingot and (**b**) plate.

**Figure 6 materials-14-05703-f006:**
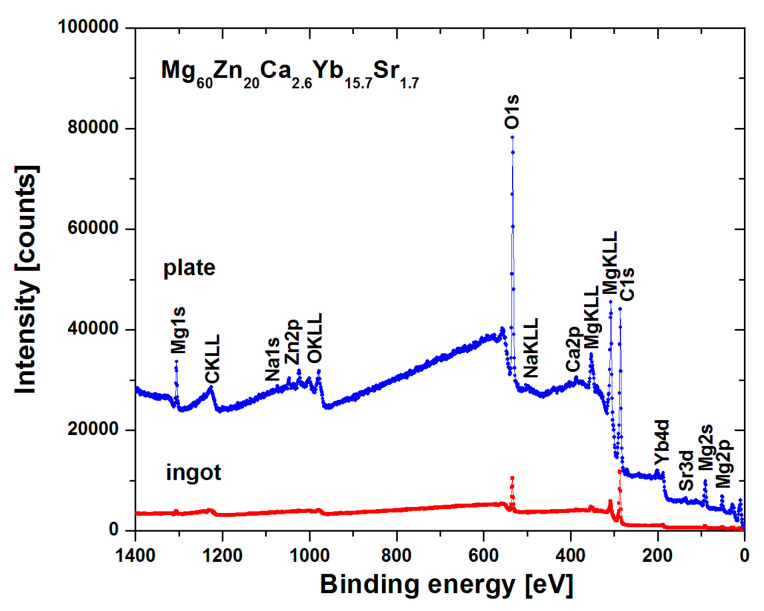
XPS survey spectra of Mg_60_Zn_20_Ca_2.6_Yb_15.7_Sr_1.7_ alloy in a form of ingots and plates after corrosion tests in Ringer’s solution at 37 °C.

**Figure 7 materials-14-05703-f007:**
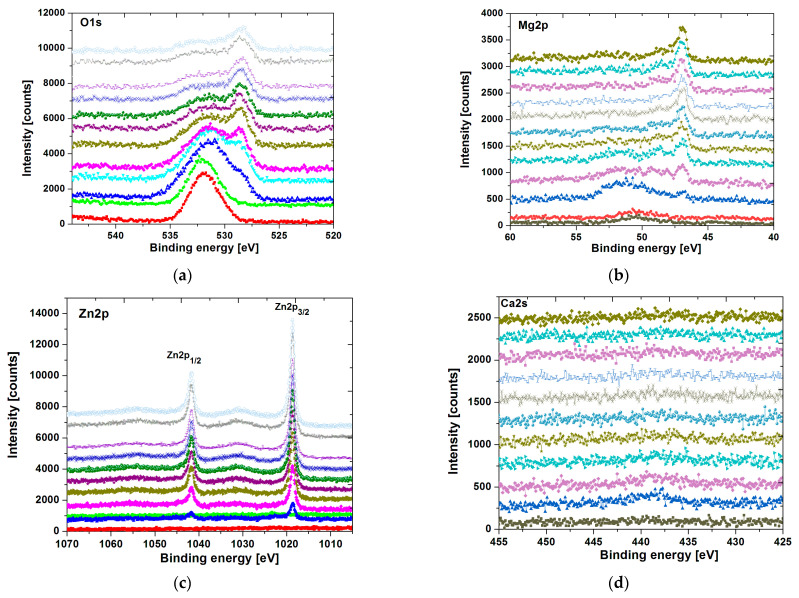
XPS core level lines of O1s (**a**), Mg2p (**b**), Zn2p (**c**), Ca2s (**d**), Yb4d (**e**), and Sr3d (**f**) of Mg_60_Zn_20_Ca_2.6_Yb_15.7_Sr_1.7_ ingot after corrosion tests in Ringer’s solution at 37 °C.

**Figure 8 materials-14-05703-f008:**
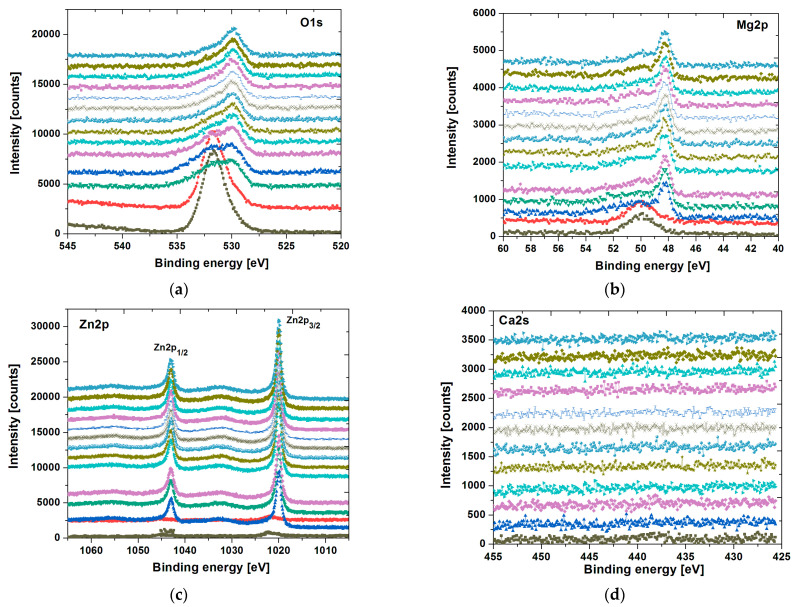
XPS core level lines of O1s (**a**), Mg2p (**b**), Zn2p (**c**), Ca2s (**d**), Yb4d (**e**), and Sr3d (**f**) of Mg_60_Zn_20_Ca_2.6_Yb_15.7_Sr_1.7_ plate after corrosion tests in Ringer’s solution at 37 °C.

**Figure 9 materials-14-05703-f009:**
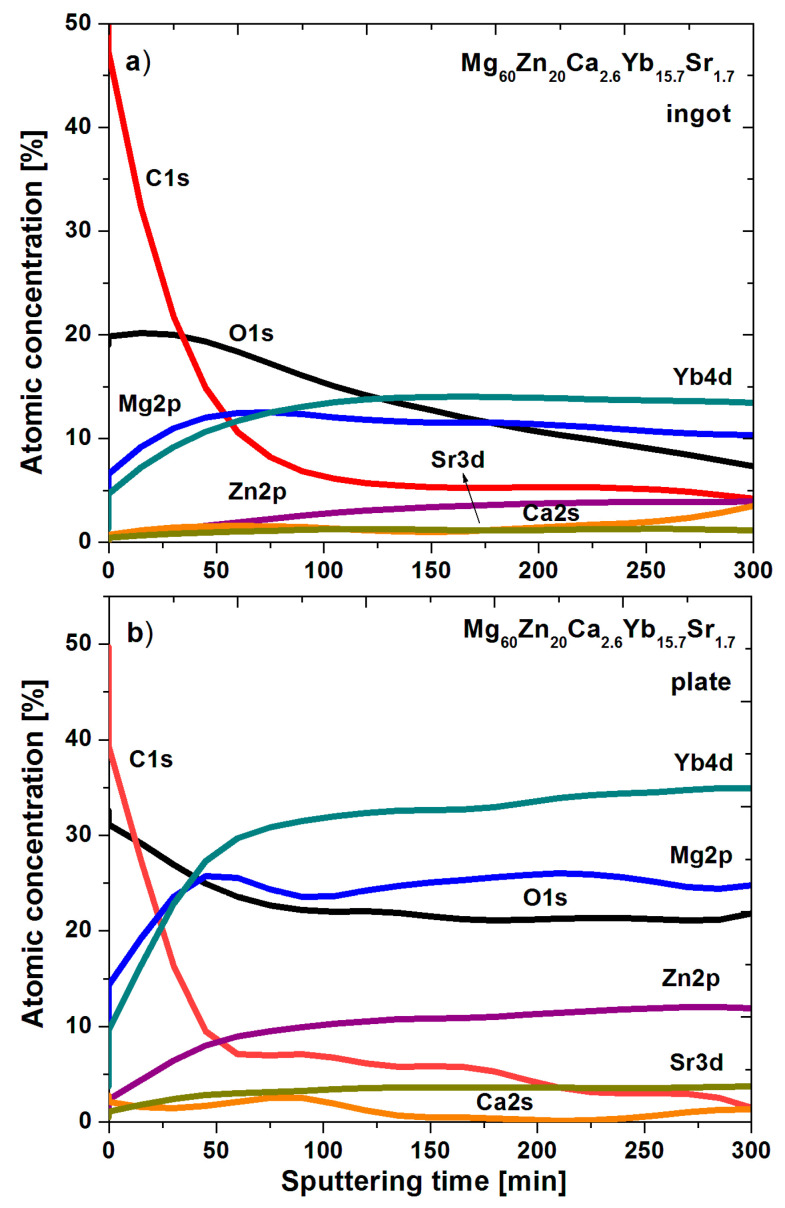
DP-XPS result for Mg_60_Zn_20_Ca_2.6_Yb_15.7_Sr_1.7_ alloy in a form of ingot (**a**) and plate (**b**) after corrosion tests in Ringer’s solution at 37 °C.

**Table 1 materials-14-05703-t001:** Results of electrochemical tests of Mg_60_Zn_20_Ca_2.6_Yb_15.7_Sr_1.7_ alloy in a form of ingots and plates (E_OCP_—open circuit potential, E_corr_—corrosion potential, β_a_, β_c_—anodic and cathodic Tafel slopes, R_p_—polarization resistance, j_corr_—corrosion current density).

Sample		E_OCP_ [V]	E_corr_ [V]	|β_a_| [mV/dec]	|β_c_| [mV/dec]	R_p_ [Ω]	j_corr_ [µA/cm^2^]
ingot	Mg_60_Zn_20_Ca_2.6_Yb_15.7_Sr_1.7_	−1.45	−1.42	1180	541	40.72	3961.8
plates	Mg_60_Zn_20_Ca_2.6_Yb_15.7_Sr_1.7_	−1.39	−1.37	90	236	139.01	203.45

## Data Availability

Data sharing is not applicable to this article.

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
