# Peer review of "The Influence of Rapid Solidification on Corrosion Behavior of Mg60Zn20Yb15.7Ca2.6Sr1.7 Alloy for Medical Applications"

_materials, 2021, doi:10.3390/ma14195703_

Round 1

Reviewer 1 Report

The presented to Materials journal manuscript 1367934 describes the influence of Mg60Zn20Yb15.7Ca2.6Sr1.7 alloy solidification on its corrosion behavior. The alloy microstructure as well as its crystallization and corrosion products were studied. It was found that the alloy produced at higher cooling rate is more resistive to corrosion than as prepared by inductive melting Mg alloy with the same additives.

The manuscript is well structures and clearly describes the state of the art. A good comparison of the obtained results and discussion is made with the already published data. Moreover, new insights into Mg-Zn-Yb-Ca-Sr alloys corrosion process were made, which would have an impact on the development of biodegradable materials for medical application.

I would recommend this manuscript to be published in Materials journal after minor corrections. Here they are.

Line 20. Please, decode DSC.

Lines 36 and 38. Need references.

Line 71. Remove “many”.

Line 90. Should be “were investigated”

Line 120. How were the master alloy ingots cooled/produced? Gravity casting?

Correct please : “…were re-melted and casted in the form of plate by HPDC…”

Line 126. Please, indicate which SEM equipment model was used?

Lines 135 and 136. Please, put information about the potentiostat, i.e. “(Metrohm AG…)” right after “Autolab 302N potentiostat”. Otherwise it is seemed that there were 2 potentiostats used.

Line 159. Add references to your images: “... by dendritic structure (fig.1 a and b).”

Line 164. The same as in line 159: “alloying elements (fig. 1 c).”

Line 167. Add a note about EDS area, namely “…EDS spectra of selected area (frame 1)”.

Line 179-180. There should be figure 2 in stead of figure 3, probably. If so, please, change also the numeration of this and others pictures in the whole manuscript.

Line 171. Please, indicate the number of figure when describing the results. Should be Fig.2.

Line 174. Need to indicate in which figure is that what you described. Should be Fig. 3.

Line 186. Should be “consists”.

Line 188. One closing “)” is missing.

Line 190. Need a reference to a literature.

Line 193. Replace “were” by “are”.

Line 198. Add a reference to the table 1. “…corrosion current density (Table 1).”

Line 207. “… showed a significant improvement in relation to pure Mg”. From where it is observed? There is no data about pure Mg in the table 1. If you take Ecor from the literature, please specify its value in the manuscript.

Line 208. “...showed more negative value” compared to what? Need a reference.

Line 208-210. Correct the phrase to the following: “However, in this article more favourable corrosion current density values were achieved, namely, 21.38 mAcm-2 and 12 12 mAcm-2 for Mg60Zn35Ca5 in the form of CIP compacted pellet consisting of amorphous structure and the sintered pellet respectively [30].” Check the value of 12 12 mAcm-2. Is it 12.12 or 12 12?

Line 212. What does MAO mean? Need a reference in the end of the phrase.

Lines 210 215. Which “another article”? put a reference to it, please.

Line 225. Should be “After the electrochemical test the samples were observed by stereoscopic microscope”.

Lines 251, 254, 257, 261 – check the spelling of some words.

Line 261-262. Verify the phrase. There should be 2 sentences, not 1.

Line 293. Remove one of two words “surface”

Line 303. A word is missing: “…may be related to different chemical ???”.

Good work!

Reviewer 2 Report

Article: The influence of rapid solidification on corrosion behavior of Mg60Zn20Yb15.7Ca2.6Sr1.7 alloy for medical applications present, in a poor way, many interesting results but with many problems of presentation. 

Starting from the title I am wondering about the repeatability of the alloy obtaining, how the authors propose to realize this action using induction melting method. 

L19: explain BSED, EDS and DSC

L25: re-phrase 

L41 : a reference is required after 12 to  24 weeks 

Introduction section must be better restructured 

L91:re-phrase 

L126: mention the type of sem and eds 

L127: dsc model used 

L167: explanation for a,b and c , in fig 1 c) mention the Y ax unit 

Where is figure 2 ???

combine figure 3 and 4 , why do you have one heat flow in a.u. fig 3 and one in w/g - figure 4 ???

L185: re-phrase 

L312: more comments on the values of micro-hardness 

In section 2 mention the equipment and method for hardness determinations 

All the conclusions must be re-written   

Round 2

Reviewer 2 Report

L128: Why do you present the EBSD detector in structural investigations and no EBSD results in experimental section? please clarify   
